# Measuring quality of family planning counselling and its effects on uptake of contraceptives in public health facilities in Uttar Pradesh, India: A cross-sectional analysis

**Arnab K. Dey**[1,2¤a]*, **Sarah Averbach**[1¤a], **Anvita Dixit**[1,2¤a], **Amit Chakraverty**[3],
**Nabamallika Dehingia**[1,2¤a], **Dharmendra Chandurkar**[3¤b], **Kultar Singh**[3¤b],
**Vikas Choudhry**[3¤b], **Jay G. Silverman**[1¤a], **Anita Raj**[1¤a]

**1** Division of Global Public Health, Center on Gender Equity and Health, University of California, San Diego
School of Medicine, La Jolla, CA, United States of America, **2** Joint Doctoral Program, San Diego State
University/University of California San Diego, San Diego, CA, United States of America, **3** Sambodhi
Research and Communications, Noida, Uttar Pradesh, India

¤a Current address: Central Research Services Facility (CSRF), La Jolla, CA, United States of America
¤b Current address: C-126, C Block, Sector 2, Noida, Uttar Pradesh, India
* akdey@health.ucsd.edu

Australia, AUSTRALIA

**Data Availability Statement:** The dataset
supporting the conclusions of this article is

## Abstract

### Background

Quality of care in family planning traditionally focuses on promoting awareness of the broad
array of contraceptive options rather than on the quality of interpersonal communication
offered by family planning (FP) providers. There is a growing emphasis on person-centered
contraceptive counselling, care that is respectful and focuses on meeting the reproductive
needs of a couple, rather than fertility regulation. Despite the increasing global focus on per-
son-centered care, little is known about the quality of FP care provided in low- and middle-
income countries like India.

This study involves the development and psychometric testing of a Quality of Family
Planning Counselling (QFPC) measure, and assessment of its associations with contracep-
tives selected by clients subsequently.

### Methods

We analyzed cross-sectional survey data from N = 237 women following their FP counsel-
ling in 120 public health facilities (District Hospitals and Community Health Centers) sam-
pled across the state of Uttar Pradesh in India. The study captured QFPC, contraceptives
selected by clients post-counselling, as well as client and provider characteristics. Based on
formative research and using Principal Component Analysis, we developed a 13-item mea-
sure of quality of FP counselling. We used adjusted regression models to assess the associ-
ation between QFPC and contraceptive selected post-counselling.

available in the Harvard Dataverse (https://doi.org/10.7910/DVN/98VZV7).

**Funding:** This study was funded by the Bill and Melinda Gates Foundation [Grant Nos. OPP1083531 and INV-010695/OPP1163682]. Sambodhi Research and Communications provided support in the form of salaries for authors AKD, AC, ND, DC, KS, and VC, but did not have any additional role in the study design, data collection and analysis, decision to publish, or preparation of the manuscript. The specific roles of these authors are articulated in the 'author contributions' section.

**Competing interests:** The author(s) declare that they have no competing interests. The study was funded by the Bill and Melinda Gates Foundation. Authors AKD, AC, ND, DC, KS, and VC were employed by Sambodhi Research and Communications. This does not alter our adherence to PLOS ONE policies on sharing data and materials.

**Abbreviations:** ANM, Auxiliary Nurse Midwives; ANOVA, Analysis of Variance; aRR, Adjusted Risk Ratio; CHC, Community Health Center; CI, Confidence Interval; DH, District Hospital; FP, Family Planning; HMSC, Health Ministry Screening Committee; ICMR, Indian Council for Medical Research; IUD, Intrauterine Device; LARC, Long Acting Reversible Contraceptives; PCA, Principal Component Analysis; QFPC, Quality of Family Planning Counselling; Std. Dev., Standard Deviation; UP, Uttar Pradesh; US, United States.

## Results

The QFPC measure demonstrated good internal reliability (Cronbach alpha = 0.80) as well as criterion validity, as indicated by client reports of high QFPC being significantly more likely for clients with trained versus untrained counsellors. We found that each point increase in QFPC, including increasing quality of counselling, is associated with higher odds of clients selecting an intrauterine device (IUD) (aRR:1.03; 95% CI:1.01–1.05) and sterilization (aRR:1.06; 95% CI:1.03–1.08), compared to no method selected.

## Conclusions

High-quality FP counselling is associated with clients subsequently selecting more effective contraceptives, including IUD and sterilization, in India. High-quality counselling is also more likely among FP-trained providers, highlighting the need for focused training and monitoring of quality care.

## Trial registration

CTRI/2015/09/006219. Registered 28 September 2015

## Background

Family Planning (FP) supports the health and well-being of women and children globally [1, 2]. Use of modern reversible contraception has been shown to prevent unintended pregnancy [3] and short inter-pregnancy intervals, both of which lead to adverse health consequences for mothers and infants [4–8]. While there are several facilitators of contraceptive use, high-quality interpersonal communication from FP providers including counselling on proper use and side-effects [9–14], clarification of misconceptions [15, 16], and addressing spousal dynamics like covert use and couple communication [17–19] are associated with contraceptive uptake and continuation among women [20].

Efforts to bring the needs and rights of patients to the center was laid out way back in 1994 at the International Conference on Population and Development (ICPD) in Cairo [21], followed by the 2001 Institute of Medicine (IOM) report that outlined patient centered care as one of the 6 goals to improve healthcare [22]. Despite these early advances, focus on providing patient centered quality of care remained low. However, there has been a renewed focus on moving beyond the traditional approach of just focusing on promoting awareness around the array of contraceptive options according to tiered efficacy, towards a more patient-centered contraceptive counselling approach, that is respectful and focuses on aspects related to the quality of interactions between providers and patients [23]. This renewed focus on patient-centered care highlights the urgent need to move away from contraceptive target goals as metrics of success and focus on development of measures that are consistent with this focus on patient centered care.

There has been theoretical conceptualization to understand components of QoC in FP counselling [23–27]. In a landmark study Bruce [24] described measuring quality "in terms of potential demographic impact" . . . fails to value the "interpersonal dimensions of care." She described six qualities of FP counselling that should be measured, including choice of methods, technical competency, provision of information, management of side effects, follow up care, and integration with other reproductive health services. Since then, this framework has

been revised by Jain and Hardee [27] to include promoting the safe provision of contraceptive technologies, provision of information in a two-way exchange, follow up care which includes guidance on switching methods (or discontinuation of a method), interpersonal relations which focus on dignity, respect, privacy, and confidentiality [27]. Complementary to the FP quality of care framework, Huezo and Diaz [28] brought attention to the need for providers to focus on meeting their clients' reproductive needs and recommended that training of providers is essential for them to be able to provide quality of care to their clients.

Key to the value of this approach is its measurement in practice. Although there has been some work around the development and validation of measures around quality of care in FP [29, 30], research assessing these multiple dimensions of quality of care and testing its association with contraceptive use in low- and middle- income countries like India is limited.

In India, about 54% women report use of contraception, and this use is predominantly female sterilization which is permanent (75% of all users) [31]. The use of reversible methods (short acting e.g. pill, condom, and Long Acting Reversible Contraceptives (LARC) e.g. IUD, implants) remains low while their discontinuation is high [31]. India has recently expanded the method-mix in the public health system to include injectable contraceptives and to ensure more providers are trained to provide IUD insertions [32, 33]. The Government of India is also assessing the feasibility of introducing contraceptive implants in public health facilities [34, 35]. India has committed to international goals, including Sustainable Development Goal 3.7, to ensure universal access to family planning by 2030 [36], FP 2020, to expand services to 48 million new users of contraceptives (40% of the global 120 million target), and 100 million current users in India [37]. However, chasing ambitious global targets for FP can increase the risk for pressure from FP providers to use certain methods, which can compromise reproductive autonomy of women. This, again, emphasizes the need for a validated measure of quality of FP counselling and understanding its association with contraceptive uptake.

In this paper, we developed and validated a measure for quality of family planning counselling (QFPC) for use with women seeking FP services from public health facilities in the state of Uttar Pradesh (UP), India and assessed its association with contraceptive choice among clients. UP is India's most populous state which has very low rates of modern contraceptive use [38]. There is shortage of healthcare providers in the state [39, 40], and a large proportion of the population dependent on the public health system to access FP methods.

Aligning with the Bruce framework [24], our measure includes information on assessing women's reproductive goals, exchange of information between counsellor and provider, and respectful interpersonal interaction. Findings from this work can offer measurement guidance for further research and our measure can be used in practice to help ensure quality of care FP counselling at a time of rapidly escalating targets and service provision in the country.

## Materials and methods

### Study setting

We analysed survey data from a study on Quality of Care in Family Planning services in UP, the most populous state in India with a population of more than 200 million. The state of UP has low contraceptive utilization, with only 29.3% of women using any modern method of contraceptive, of which close to 60 percent is Female Sterilization [31]. The state relies on its health system to provide access to FP services, with public health facilities reported to be the main source of accessing modern contraceptive methods for more than half the people (54.1%) in the state [38]. The study, conducted between December 2016 and February 2017 in 120 public health facilities, involved surveys with women receiving FP counselling, client's interpersonal experience with the counsellor, readiness of facilities to provide FP services,

knowledge and skills of providers to provide FP services, and providers' adherence to clinical protocols during service provision.

## Sampling procedure

Facilities for the study were sampled from public health facilities in the state that provided mini-laparotomy sterilization and IUD services. The sampling frame was obtained from the Health Management Information System (HMIS) data pertaining to the previous Indian financial year (April 2015 to March 2016). The list of facilities in the sampling frame was limited to District Hospitals (DHs) and Community Health Centres (CHCs) at the block level that provided mini-laparotomy sterilization and Intrauterine Device (IUD) insertion services in the previous financial year. We restricted inclusion to these facilities to allow for assessment of FP services with sterilization and IUD available on site. The sampling frame, thus obtained, included 178 facilities from 75 districts in the state. Finally, 120 facilities were sampled from the sampling frame, using probability proportionate to size (PPS) sampling based on the number of sterilization procedures conducted in the facility in the previous financial year. This number was chosen as a proxy to the client load in the facilities. Of these 120 facilities, 50 were DHs and the remaining 70 were CHCs. The research teams stayed at each of the facilities for a period of 3–5 days to observe clinical practices around sterilization and IUD insertion. During their stay at the facility, research staff invited all women who visited the facility for FP counselling to participate in the study. Of a total of 289 women who were thus invited, N = 237 women agreed to participate. The design is further described elsewhere [41].

## Data collection

Female nurses trained in survey data collection, including ethical treatment of respondents for research and elicitation of sensitive information, served as our research interviewers for this study. These nurses were not affiliated with any public health facilities in the period of the study. Interviewers approached all women who had received counselling at the facility and asked if they would like to participate in a brief survey to share their experiences from the counselling session. Participants provided written informed consent before the interview and received no monetary incentive to participate in the study. The research staff interviewed participants in a private setting using handheld mobile devices. The interview included questions on aspects of quality of counselling received and the method selected by participants after the counselling session.

## Ethics

Institutional review board (IRB) approval for this study was granted from Public Health Service- Ethical Review Board (PHS-ERB) and from the Health Ministry Screening Committee (HMSC) facilitated by Indian Council for Medical Research (ICMR). IRB review and approval for the current analyses was obtained from human research protections program at the University of California, San Diego.

## Measures

**Dependent variable.** The dependent variable used in the study was the FP method selected by women post counselling. We categorized responses as female sterilization, intra-uterine device (IUD), short-acting methods (condoms and oral contraceptive pills) and, no preferred method.

**Independent variables.** Our primary independent variable of interest was the *Quality of FP Counselling (QFPC)* scale—a measure developed for this survey. The thirteen items for this measure were developed based on expert input and literature review, as well as prior qualitative research on women's and providers' experiences with FP counselling. Cognitive testing was undertaken for the survey including these items with the population of focus and providers working with them, to ensure clarity of items for potential respondents. As our primary interest was specific to the patient-provider interaction during counselling, we used the following three elements of the Bruce [24] framework a) FP counsellors' provision of information b) eliciting client's FP history and preferences, and c) the respectful and engaging interaction between the counsellor and the client. The first element–*provision of information* included items on providers informing clients about different contraceptives, explaining method use, explaining possible side-effects and advising them on what to do in case they face problems. Based on extensive research, experiencing side-effects was identified as a primary reason for contraceptive discontinuation [42], so special emphasis was made to include items on this issue. The second element–*eliciting client's preferences* included items on providers asking clients about their fertility goals, different FP methods used earlier, problems faced with methods used earlier and clients' preferred method of choice. The third element—*respectful and engaging interaction* included items on clients being treated in a respectful and friendly manner by providers, providers spending sufficient time with them during the session and providers not applying any pressure to select a particular contraceptive (as reported by clients). All items were assessed with a yes or no.

Our secondary independent variable training of providers on FP counselling. Training of providers was used as a dichotomous variable coded as 1 if providers had received specific training on FP counselling and 0 otherwise.

**Client level covariates.** Our survey captured socio-demographic characteristics of participants including client's age, caste, religion, education, number of living children, presence a male child and prior use of modern FP method by clients. Age of the client was used as a continuous variable in the survey. Caste was used as a categorical variable coded as Scheduled Caste/ Scheduled Tribe, Other Backward Classes (OBC) and General category. Religion was coded as a dichotomous variable classifying clients as Muslim and Non-Muslim. The choice of classifying religion of clients into Muslim and non-Muslim was made in accordance with the disproportionately low use of modern methods of contraceptives by Muslim couples in the state, relative to couples from other religions [38]. Educational attainment of clients was also used as a dichotomous variable coded as those who have completed at least primary education and those who have not. Number of living children was used as a categorical variable identifying couples who had one child, two children and three or more children. Presence of a male child was used as a dichotomous variable coded 1 if the client had any male child and 0 if not. This was done in accordance to the state level trend of lower use of modern contraceptive methods among couples who do not have any male children [38]. Prior use of modern FP method by clients was used as a dichotomous variable and was coded 1 to identify clients who had used any modern FP method before and 0 otherwise.

**Provider level covariates.** We also captured provider characteristics via structured interviews with providers who were provided counselling services to clients on FP. These included age, gender, designation, and previous training received on FP counselling. Provider age was used as a continuous variable. Gender of providers was used as a dichotomous variable identifying male and female providers. Provider designation was also treated as a dichotomous variable indicating if the providers were designated FP counsellors in the facility or whether they were staff-nurses or Auxiliary Nurse Midwives (ANMs). Provider data were linked with client

data using unique identifiers assigned to both providers and clients at the time of initiating the survey at a facility.

## Data analysis

We assessed internal reliability of our QFPC measure using Cronbach alpha, and we assessed construct validity using Principal Component Analysis (PCA). Since the items in our measure were not standardized, the correlation matrix was used to extract the components [43]. Kaiser's criterion was used to retain the components with an Eigen value of more than 1.0 [44–46]. Varimax rotation was used to obtain the proportion of variance in the data explained by each of the retained components. Subsequent to extraction of the components, the proportion of variance explained by each component was used to generate weights for each of the components and was further multiplied by the predicted scores of each of the components and added together to obtain a composite index of quality [45]. The composite score was then normalized to range from 0 to 100 to generate the QFPC measure.

To assess criterion validity of QFPC, we used multivariable linear regression to test the associations between our QFPC measure and the training of providers on FP counselling.

To explore the association between contraceptive selected by the client post-counselling and the independent variables, we developed two multinomial logistic regression models. The first model (Model-1) tested the association between choice of contraceptive post-counselling and the QFPC measure. This model was adjusted for client's religion, number of living children, presence of male child, education, and prior use of modern FP method by clients and provider's age, provider's gender, provider's designation, and provider's training on FP counselling. Model-2 tested the association between FP method selected post-counselling and training of providers on FP counselling. This model was adjusted for client's religion, number of living children, presence of male child, education, and prior use of modern FP method by clients and provider's age, provider's gender, provider's designation, and type of facility. We constructed parsimonious models [47] to ensure that we did not over-adjust our analyses. For both the models, we used backward stepwise technique to create parsimonious models and ensure that we did not over adjust our analysis. Analyses were conducted using R version 4.0.2.

## Results

### Client characteristics

Participant's age ranged from 19 to 42 years (mean age = 27.51, Std. dev. = 4.25) (see Table 1). Majority of the participants were Non-Muslims (89.87%) and 21.94% of the participants belonged to socially marginalized groups (Scheduled Caste or Scheduled Tribe). Almost half of the participants (50.21%) had 3 or more living children and majority of them had at least one male child (86.50%). Close to two-thirds of the participants (64.98%) had completed primary education at the time of the survey and a similar proportion reported that they had never used a modern FP method before (64.14) (Table 1).

### Contraceptive selected by clients post-counselling

Clients were asked about the method that they selected after counselling. Almost half of the participants (41.77%) reported that they selected female sterilization, followed by a quarter of the respondents (25.74%) reporting they chose IUDs and one-fifth of participants (19.41%) reporting they selected a short-acting method (oral contraceptive pills or condoms). Thirteen

**Table 1. Descriptive statistics of contraceptive method preferred post-counselling and key client level covariates (N = 237).**

| | | n | % or Mean (Std. Dev.) |
|---|---|---|---|
| **Dependent Variable** | | | |
| **Method Preference post-Counselling** | No Method | 31 | 13.08 |
| | Short-acting methods | 46 | 19.41 |
| | IUD | 61 | 25.74 |
| | Female Sterilization | 99 | 41.77 |
| **Independent Variable** | | | |
| Quality of FP Counselling (QFPC) | Mean (Std. Dev) | 237 | 70.33 (24.43) |
| **Individual level covariates** | | | |
| Caste | SC/ST | 52 | 21.94 |
| | OBC | 150 | 63.29 |
| | General | 35 | 14.77 |
| Religion | Muslim | 24 | 10.13 |
| | Non-Muslim | 213 | 89.87 |
| Age of women | Mean (Std. Dev) | 237 | 27.51 (4.25) |
| No. of living children | 1 living child | 41 | 17.30 |
| | 2 living children | 77 | 32.49 |
| | 3 or more | 119 | 50.21 |
| Male Child | Yes | 205 | 86.50 |
| | No | 32 | 13.50 |
| Completed Primary education | Yes | 154 | 64.98 |
| | No | 83 | 35.02 |
| Prior use of modern FP method | Yes | 85 | 35.86s |
| | No | 152 | 64.14 |

percent of the respondents (13.08%) also reported that they did not have a preferred method post-counselling (Table 1).

## Provider characteristics

A total of 144 healthcare providers counselled the women in the study sample. Health providers offering FP counselling were aged 21 to 59 years (mean age = 34.42 years, std. dev. = 9.21) and most were women (89.58%). A little less than half of the providers were designated FP counsellors (40.28%) while the remaining 59.72% of the providers comprised of staff-nurses and Auxiliary Nurse Midwives (ANMs) acting as FP counsellors. A little over one-third of the providers (38.89%%) had not received any training specific to FP counselling (Table 2).

**Table 2. Descriptive statistics of provider level characteristics (N = 144).**

| | | N | % or Mean (Std. Dev.) |
|---|---|---|---|
| Age of provider | Mean (Std. Dev) | 144 | 34.42 (9.21) |
| Gender of provider | Male | 15 | 10.42 |
| | Female | 129 | 89.58 |
| Provider Designation | FP Counsellor | 58 | 40.28 |
| | Staff-Nurse / ANM | 86 | 59.72 |
| Trained on Counselling | Yes | 88 | 61.11 |
| | No | 56 | 38.89 |

## Psychometric analysis of the QFPC measure

We used Principal Component Analysis (PCA) to test QFPC for construct validity. PCA generated 4 components with an Eigen value of more than 1.0.

The first component explained 34 percent of the total variation and represented the sharing of information between providers and clients: providers eliciting information from clients about their preference and experience with FP methods in the past and sharing information about FP methods suited to these preferences. We interpret this component as the exchange of information between the provider and the client, and elicitation of client preferences. The second component explained 13 percent of the total variance and showed strong positive loadings for participants reporting to have been treated in a friendly and respectful manner. We interpret this component as respectful interaction by the provider during the counselling session. The third component explained 11 percent of the total variation in the data and had strong positive loadings for participants reporting that they were encouraged to ask questions and that providers spent sufficient time with them during counselling. We interpret this as creation of an environment by the provider that is supportive of client's autonomy. The fourth component explained 8 percent of the variance and had a strong positive loading for participants reporting that they did not feel pressured by providers to select any specific FP method (S1 Appendix). The mean score of the overall Quality of Family Planning (QFPC) measure received by clients was 70.33 out of 100 (std. dev. = 24.43) (Table 1).

We used Cronbach alpha to test for internal reliability for our 13 item QFPC measure, and found good internal reliability for this measure (Cronbach alpha = 0.80) (Table 3).

## Criterion validity

To assess criterion validity, we examined the associations between our QFPC measure and training of providers on FP counselling, using multivariable linear regressions. We found that quality of counselling was positively associated with training of providers specific to FP counselling (adj. coef. = 6.73, 95% CI: 2.18–11.29) (Table 4).

**Table 3. Individual items used to assess Quality of FP Counselling (QFPC) (N = 237).**

| Individual Items to assess Quality of Counselling on Family Planning | n | % |
|---|---|---|
| Did the provider ask you about your reproductive goal, i.e. how many children do you have, how many you want? | 150 | 63.29 |
| Did the provider ask you about different methods you have used earlier? | 139 | 58.65 |
| Did the provider ask you about problems you have had with earlier methods? | 115 | 48.52 |
| Did the provider ask your method preference? | 137 | 57.81 |
| Did the provider tell you about different FP methods? | 141 | 59.49 |
| Did the provider explain you how to use the method you selected? | 139 | 58.65 |
| Did the provider tell you about possible side effects of the method you selected? | 110 | 46.41 |
| Did the provider tell you what to do if you experience any problem after using the method you selected? | 146 | 61.60 |
| Did the provider encourage you to ask questions? | 177 | 74.68 |
| Was the time spent in consultation sufficient to discuss your needs? | 215 | 90.72 |
| Did the provider treat you in a friendly way? | 209 | 88.19 |
| Did provider treat you in a respectful way? | 219 | 92.41 |
| Anytime during the discussion with the health provider, did you feel that he/she is pressurizing you to select a particular family planning method? | 171 | 72.15 |
| **Cronbach Alpha** | | **0.80** |

**Table 4. Adjusted linear regression to test the association between Quality of FP Counselling (QFPC) and training of providers on FP counselling characteristics (N = 237).**

|  |  | Adjusted Coefficient | 95% LCI | 95% UCI |
|---|---|---|---|---|
| Provider trained on FP counselling [a] | No | Ref | - | - |
|  | Yes | 6.73 | 2.18 | 11.29 |

[a] Model adjusted for provider age, provider designation, type of facility and client's caste.

## Associations between FP method selected post-counselling and QCFP

Multinomial regression adjusted for client and provider level covariates (Model-1) shows that for each point increase in QFPC score the participants are more likely to select short-term methods (ARRR:1.02, 95% CI: 1.00–1.05) IUD (ARRR:1.03; 95% CI:1.00–1.05) and female sterilization (ARRR:1.06; 95% CI:1.03–1.08) as compared to choosing no FP method post-counselling (Table 5).

## Association between FP method selected post-counselling and training of providers on FP counselling

We also found that type of FP method selected by clients post counselling was associated with providers receiving previous training specific to FP counselling (Model-2). Clients who were counselled by providers previously trained in FP counselling were more likely to select IUDs (ARRR: 8.20, 95% CI: 2.67–25.11) and female sterilization (ARRR:4.52, 95% CI: 1.62–12.56), as compared to choosing no FP method post-counselling (Table 5).

**Table 5. Multinomial logistic regression models to test the association between type of FP method selected post-counselling and a) Quality of FP Counselling (QFPC) and b) training of providers on FP counselling (N = 237).**

|  |  | ARRR | 95% LCI | 95% UCI |
|---|---|---|---|---|
| **No method selected post-counselling** |  | Base Outcome | | |
| **Short-acting methods** |  |  |  |  |
| Quality of FP Counselling (QFPC) [a] |  | 1.02 | 1.00 | 1.05 |
| Provider trained on counselling [b] | No | Ref | - | - |
|  | Yes | 1.27 | 0.42 | 3.78 |
| **Intra-Uterine Devices** |  |  |  |  |
| Quality of FP Counselling (QFPC) [a] |  | 1.03 | 1.01 | 1.05 |
| Provider trained on counselling [b] | No | Ref | - | - |
|  | Yes | 8.20 | 2.67 | 25.11 |
| **Female Sterilization** |  |  |  |  |
| Quality of FP Counselling (QFPC) [a] |  | 1.06 | 1.03 | 1.08 |
| Provider trained on counselling [b] | No | Ref | - | - |
|  | Yes | 4.52 | 1.62 | 12.56 |

[a] Model adjusted for client's religion, number of living children, presence of male child, education, and prior use of modern family planning method by clients and provider's age, provider's sex, provider's designation, and provider's training on FP counselling.

[b] Model adjusted for client's religion, number of living children, presence of male child, education, and prior use of modern family planning method by clients and provider's age, provider's sex, provider's designation and type of facility.

## Discussion

Findings from the study demonstrate the reliability and validity of the Quality of FP Counselling (QFPC) measure for use among women seeking FP counselling in India. The QFPC measure posits exchange of information between providers and clients, friendly and respectful interaction, supportive environment created by the provider and no pressure to uptake a method as important dimensions of quality of FP counselling. This is closely aligned with elements of the Quality of Care framework recommended by Bruce [24], especially–a) FP counsellors' provision of information b) elicitation of client's family planning history and preferences, and c) the respectful and engaging interaction between the counsellor and the client.

Findings are also aligned with recent studies that tested measures on Quality of Care in multiple settings. Sudhinaraset et al. [48] developed the Person-Centered Family Planning Scale in India and Kenya and identified two subscales related to "autonomy, respectful care, and communication" and "health facility environment" to be relevant in both the contexts. Holt et al. [49] developed the Quality of Contraceptive Counselling (QCC) Scale, in Mexico, and identified 1) information exchange, 2) interpersonal relationship, and 3) disrespect and abuse as the underlying dimensions of quality of contraceptive counselling. Jain et al. [50] used a similar approach to develop and validate a contraceptive care measure and identified 1) respectful care, 2) method selection, 3) effective use of method selected and 4) continuity of contraceptive use and care as the four underlying domains of quality. More recently, Johns et al. adapted the Interpersonal Quality of Family Planning (IQFP) scale [29] and validated it in the Indian context [30]. The 11 item IQFP scale also includes several items similar to the QFPC measure. While there are several measures around Quality of Contraceptive Counselling that have been developed and tested, our QFPC measure has been tested in the Indian context, has lesser items than most of the other scales and includes binary response patterns (yes/no) which makes it easier to administer in different contexts. This makes the QFPC measure a useful tool to measure patient-centered FP counseling in low- and middle- income settings.

Our study also suggests that the quality of counselling is positively associated with providers being previously trained on FP counselling. This is especially concerning given that more than a third of the providers in the sampled facilities had not received training on FP counselling. This may be indicative of a shortage of staff in general as well as those dedicated to FP counselling. Severe shortages of staff in public health facilities result in facilities being unable to spare providers to attend trainings organized by the Government, lest the facility will have no / few providers left to provide healthcare services during the course of the training. While there is limited literature addressing the lack of dedicated FP counsellors in India, the general gap in the availability of health workforce in the country and its adverse effects is well documented in prior research [51, 52] and appears to extend to FP counselling as well.

We also found that the choice of contraceptive post-counselling was associated with the quality of counselling received by women. Better quality of counselling was associated with higher uptake of short-term methods, IUDs and female sterilization relative to choosing no method at all. Choice of contraceptive was also found to be associated with training of providers on FP counselling. Providers who were trained on FP counselling were more likely to have clients who opt for IUDs and female sterilization after counselling relative to choosing no method at all. This is especially relevant to the Indian context where the distribution of FP users is highly skewed towards sterilization and only 3% of women using modern contraceptives use long acting reversible contraceptives (LARCs) such as IUDs, which are more effective than other forms of reversible contraceptives such as pill or condom [31]. Our study indicates

that training of providers specific to FP may facilitate capacities to engage with clients on a broader array of contraceptive options, and thus improve uptake of more effective reversible forms of contraceptives. These findings highlight the value of investing in filling the HR gaps and training of providers to provide comprehensive contraceptive counselling, as it has potential to positively affect the health of mothers and infants by increasing the pregnancy intervals and reducing unintended pregnancies [7, 8].

Overall, these findings highlight the utility of the QFPC measure in assessing delivery of patient-centered care. Our findings also underscore the need for enhancing both the quality and quantity of trainings for providers to deliver the multidimensional elements of quality FP counselling in UP, though importantly, they are achieving supportive and respectful care. Nonetheless, the value of training specific to FP counselling cannot be understated given its association with higher quality counselling and client's preference for more effective contraceptives post-counselling. Given the low prevalence of IUD use in the country [31], these findings support the value of high-quality person-centered FP counselling to help broaden the array of spacing contraceptives used in India.

While findings are promising regarding the value of high-quality FP counselling as well as the standard delivery of respectful care in these settings, it must be noted that a not small percent of women coming in for FP counselling leave with a preference for no contraceptive. Current findings indicate that this may be, at least in part, attributable to poor quality counselling, based on the findings of a negative association between quality counselling and preference for no contraceptives. Another reason for women not selecting any method post-counselling could be the receipt of counselling by untrained providers as evident from the association between counselling by trained providers and choice of more efficient contraceptive methods. This further substantiates the need to ensure that every facility providing FP services should have providers trained on FP and measurement of patient-centered quality of care be prioritized using tools like the QFPC. Further research is needed to understand better why women opt to leave with no contraceptives at all following poorer delivery of care, particularly as this sample was women who had come for FP counselling.

## Strengths and limitations

The study is based on a unique quality of care study for FP services in public health facilities in Uttar Pradesh. The study furthers the measurement discourse around quality of FP counselling and provides interesting insights into the associations between contraceptive uptake and characteristics of clients and providers.

The study is not without limitations. The unique study design and sampling approach resulted in limiting the sample to women approaching select public health facilities in the state. This makes it challenging to generalize the findings of the study. Selection bias is also a concern, as findings are limited to women presenting at public FP clinics, who may belong to a specific socio-economic background. While the study included variables on social marginalization, it did not capture information on the economic status of the participants. This may add to the challenge of generalizing the findings to all women in Uttar Pradesh or India. In addition, the study did not capture information on motivations of women to visit these facilities or whether they had received prior counselling and advise from community level health workers, both of which can act as confounding variables in the association between quality of counselling and type of contraceptive selected post-counselling. Our inability to adjust for these confounders in the analyses another limitation. Variables used in the study largely rely on self-report and thus are subject to social desirability bias. The small number of participants per clinic also limited our ability to understand clinic level differences that may contribute to

findings. Recall bias is expected to be minimal as study variables are largely indicative of preferences and counselling at the time of assessment or just preceding it. This study is cross-sectional, so causality cannot be assumed and effect of quality counselling on use and continuation cannot be assessed. Longitudinal analysis with patient follow-up would offer greater insight into uptake of contraceptives as well as continuation subsequent to counselling.

## Program implications

The study presents important implications for programs that work towards improving quality of care and uptake of contraceptives. The association of the QFPC measure with training of providers on FP counselling highlights the need to expand the number of providers who have been specifically trained on FP counselling skills. The association of contraceptive uptake post-counselling with the QFPC measure underscores the need to improve the quality with which providers interact with clients during these sessions, with special focus on dimensions of quality identified in the paper viz. information exchange, respectful interaction, supportive environment and no pressure to uptake a method.

## Conclusion

The aim of the study is to provide a new measurement of quality of FP counselling and to examine the relationship between quality counselling and contraceptive uptake. The study posits a measure for quality of FP counselling that is aligned with the Bruce framework [24] on quality of care, and finds the measure valid and reliable in the context of FP counselling in Public Health facilities in UP.

The composite measure for quality of counselling developed in the paper suggests that dimensions of information exchange, respectful interaction, supportive environment, and no pressure to uptake a method are important dimensions of quality of FP counselling. The study calls for sensitization of providers to lay special emphasis on these dimensions during counselling as quality of counselling can have a significant bearing on the contraceptive choice of women and also underscores the need for an enhanced emphasis on training more providers on topics related to FP counselling. The study also calls for further research to enable a deeper dive into the reasons for women not choosing any method at all after counselling, so that no woman seeking FP counselling for pregnancy prevention should leave the clinic without provision of a woman's choice of contraceptive.

## Supporting information

**S1 Appendix. Rotated components matrix.**
(DOCX)

## Acknowledgments

We wish to acknowledge the efforts of the survey managers and nurse investigators who implemented the study. We would also like to thank the facility leaders and providers who provided immense support to our teams undertaking the study. Finally, we are immensely grateful to the women who graciously participated in the study and patiently shared their experiences with our teams.

## Author Contributions

**Conceptualization:** Arnab K. Dey, Sarah Averbach, Amit Chakraverty, Dharmendra Chandurkar, Kultar Singh, Vikas Choudhry, Jay G. Silverman, Anita Raj.

**Data curation:** Arnab K. Dey.

**Formal analysis:** Arnab K. Dey.

**Funding acquisition:** Dharmendra Chandurkar, Kultar Singh, Jay G. Silverman, Anita Raj.

**Methodology:** Arnab K. Dey, Nabamallika Dehingia, Kultar Singh, Vikas Choudhry.

**Project administration:** Arnab K. Dey, Amit Chakraverty, Dharmendra Chandurkar, Vikas Choudhry.

**Resources:** Dharmendra Chandurkar, Kultar Singh.

**Supervision:** Arnab K. Dey, Dharmendra Chandurkar, Kultar Singh, Jay G. Silverman, Anita Raj.

**Writing – original draft:** Arnab K. Dey, Sarah Averbach, Anvita Dixit, Nabamallika Dehingia, Jay G. Silverman, Anita Raj.

**Writing – review & editing:** Arnab K. Dey, Sarah Averbach, Anvita Dixit, Jay G. Silverman, Anita Raj.

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
