## [Decision Letter · Decision Letter 0]

26 Nov 2020

PONE-D-20-28620

Measuring quality of family planning counseling and its effects on uptake of contraceptives in Public Health Facilities in Uttar Pradesh, India: a cross-sectional analysis

PLOS ONE

Dear Dr. Dey,

Thank you for submitting your manuscript to PLOS ONE. After careful consideration, we feel that it has merit but does not fully meet PLOS ONE’s publication criteria as it currently stands. Therefore, we invite you to submit a revised version of the manuscript that addresses the points raised during the review process.

ACADEMIC EDITOR: All five reviewers and I find that this paper has merit and can be considered if the authors address the reviewer comments. All five reviewers are well-respected scholars in the subject. Their comments range from a solid justification for the study at the introduction to giving more details on study design and to strengthen the statistical analyses and presentation of the tables. I believe many of these comments can be easily addressed. I hope to receive a revised version of this paper. 

We look forward to receiving your revised manuscript.

Kind regards,

Srinivas Goli, Ph.D.

Academic Editor

PLOS ONE

Additional Editor Comments:

All five reviewers and I find that this paper has merit and can be considered if the authors address the reviewer comments. All five reviewers are well-respected scholars in the subject. Their comments range from a solid justification for the study at the introduction to giving more details on study design and to strengthen the statistical analyses and presentation of the tables. I believe many of these comments can be easily addressed. I hope to receive a revised version of this paper.

2. Please include additional information regarding the survey or questionnaire used in the study and ensure that you have provided sufficient details that others could replicate the analyses. For instance, if you developed a questionnaire as part of this study and it is not under a copyright more restrictive than CC-BY, please include a copy, in both the original language and English, as Supporting Information.  If the original language is written in non-Latin characters, for example Amharic, Chinese, or Korean, please use a file format that ensures these characters are visible.

We note that one or more of the authors are employed by a commercial company: Sambodhi Research and Communications.

(2) Please also provide an updated Competing Interests Statement declaring this commercial affiliation along with any other relevant declarations relating to employment, consultancy, patents, products in development, or marketed products, etc.  

Please respond by return email with an updated Funding Statement and Competing Interests Statement and we will change the online submission form on your behalf.

6. We note you have included a table to which you do not refer in the text of your manuscript. Please ensure that you refer to Table 2 in your text; if accepted, production will need this reference to link the reader to the Table.

Reviewers' comments:

Reviewer's Responses to Questions

**Comments to the Author**

1. Is the manuscript technically sound, and do the data support the conclusions?

Reviewer #1: No

Reviewer #2: Partly

Reviewer #3: Partly

Reviewer #4: Yes

Reviewer #5: Partly

2. Has the statistical analysis been performed appropriately and rigorously? 

Reviewer #1: No

Reviewer #2: Yes

Reviewer #3: No

Reviewer #4: Yes

Reviewer #5: No

3. Have the authors made all data underlying the findings in their manuscript fully available?

Reviewer #1: Yes

Reviewer #2: No

Reviewer #3: Yes

Reviewer #4: Yes

Reviewer #5: Yes

4. Is the manuscript presented in an intelligible fashion and written in standard English?

Reviewer #1: Yes

Reviewer #2: Yes

Reviewer #3: Yes

Reviewer #4: Yes

Reviewer #5: Yes

5. Review Comments to the Author

Reviewer #1: The motivation of paper is interesting nut the execution is weak. The paper need to improve in presentation and implications. The limitations of the study should also be highlighted. I put few of the point for consideration

1. Introduction: It may be meaningful to begin with Quality of Care . Following this its linkages with contraceptive use. 2.Review of literature Beyond Bruce framework is needed

3. Highlight why Uttar Pradesh and study sample

4. All tables need to be scientifically presented. I suggest you give 2 column , one for % and other for N. Moreover, many of the table are conbfusing to reader. For ex: table 1 n (%) or Mean (SD)? What is ecactly presented? Is it N and % or mean and SD. there are 2 values in tables. Most of the tables arein similar fashion

5. Drop table 2

6. Move table 3 to appendix

7. OBC is not marginalised group as mentioned in text . Correct it

8. Give mean value of composite score-QOFC

9.Drop table 5. Too much segregation of small sample of about 80 is dangerous in a scientific manuscript.

10. Add Strong note on limitations . the sampe is not true for generalisation. More over you are taking a selected sample. hecne do not generalise it

11. Mention value addition of this and implications for policy

Reviewer #2: Overall, this paper has the potential to make an important contribution to the literature, particularly by pointing out the development and psychometric testing of a Quality of Family Planning Counseling. However, the sampling design and its description is not clear. The following are some of the comments.

Abstract

The abstract needs editing.

The background section of abstract need to re-write. In the background section, it is written ad “This study involves the development and psychometric testing of a Quality of Family Planning Counseling (QFPC) measure for India….. I am not sure whether this statement is correct, as this analysis use a 237 women sample from public health facilities in Uttar Pradesh and generalised for India. Even there is no clear description on sample design in method section of abstract as well as in main methods. Whether the sampled public sector facilities spread across the state o

Background

I suggest that the authors say more accurately that the purpose of the analysis in the background section. There are no specific details on the family planning services or counciling in Uttar Pradesh. It is not clearly reading the background in connection with this study objective.

Methods

Methods section is inadequate. The current method section in this manuscript reads in a very choppy way.

This section not explained how did the sample facilities selected from Uttar Pradesh, and how did the women sampled. I suggest that the authors say more accurately about the sampling design including public health facility selection, women selection etc.

Line no 123-124, it is mentioned that …. 178 facilities that met inclusion criteria, we sampled 120 facilities for study inclusion. Are these 178 facilities in the 75 districts in Uttar Pradesh or selected districts?

Line 124-125, it is mentioned that “The design is described in Mozumdar et. al. (46)” Did you use the same data and analysed for Uttar Pradesh. It would be useful to explain the sample design in this manuscript as well. This will be helpful for the readers.

Line 172-173, it is mentioned that “We additionally included measures on provider characteristics, captured through structured interviews with the specific providers who counseled women on FP services”. It would be useful to describe this clearly.

Result

Did you collect the information on usage of family planning among women. If so it is useful to show the prevalence of family planning usage.

I could understand more than 60% of FP councillors were females as most of them where ANM or Nurses. Whether any male councillors were available in these facilities?

Table 7 can modify. Please show only odds ratio and 95% CI. For example, 1.05 (1.03-1.070. P value is not required as the odds with 95% CI is sufficient for interpreting the statistical significance.

Discussion

In the discussion section, the sentence However, given the relevance of findings for public clinics, generalizability to this…………..." could be more explicit and mention clearly the reason. I am still doubtful, how the authors can generalize the result for India.

Reviewer #3: The paper titled “Measuring quality of family planning counseling and its effects on uptake of contraceptives in Public Health Facilities in Uttar Pradesh, India: a cross-sectional analysis” is a good attempt to examine the quality of contraceptive counselling and its subsequent effects on uptake of specific methods in Uttar Pradesh, India. The results are informative. Nevertheless, I suggest the followings to make the paper more suitable for publication:

Sampling design must be described in the paper, as the article is expected to stand alone; although it has been explained in other published article.

Does the survey captured the main motivator/factor that made sampled women to visit these facilities for contraception? Do these women come to the facility alone or with any family member or any grass root level health worker like ASHA/ANM? It is quite possible that these women might have been counselled for visiting these facilities for a particular method well in advance before visiting these facilities. How did the authors address this?

It is evidenced that usually most of the people seeking health care utilization from public health facilities are from low economic background. Hence inference drawn from this analysis has its own limitation so far as generalization is concerned. This may be marked in the discussion section. Does this survey gathered information about economic status of these women, if so inclusion of this in the analysis is suggested?

It is suggested to discuss in details the possible reasons for those 13% women who preferred no method post-counselling; though they visited the facility for adopting/accepting any FP method.

Why the religion was classified as Muslim and Non-Muslims? It is assumed that majority of that Non-Muslims are Hindus, who constitute a majority of the sample women. In that case why not to classify Hindus and Others.

The purpose behind capturing education up to primary level needs to be presented? Role of education may have been much better understood if it would have been collected beyond primary level.

The non-inclusion of the type of facility i.e. DH & CHC in the analysis, is unclear

Table 7: It is not very clear OR at what level significance is shown/made bold.

Were there any non-literate participant? Especially when 35% of them had not completed primary education. If so, how did the consent sought and taken from them to participate in this study.

The conclusion section must be revisited to limit it as per the analysis carried. Authors may restrict themselves from overgeneralization of the results looking at the sample size and design.

Reviewer #4: No comments at this time.

No comments at this time.

No comments at this time.

No comments at this time.

No comments at this time.

No comments at this time.

No comments at this time.

No comments at this time.

Reviewer #5: Thanks for giving me an opportunity to read this interesting paper. The authors have used a facility level exit interview data to present a topic which needs more and more research to improve quality of care for family planning services in India. While the paper has several strengths, I feel the statistical analysis is not that great. Following are observations which may help the authors in strengthening the paper.

INTRODUCTION

1. The introduction is too lengthy and I feel it can be shortened significantly. The authors should should strengthen the last but one paragraph in the introduction section (Page 6 Line 98-103) with more solid justification on why this study is called for. The current articulation is somewhat vague and may be supplemented with specific contextual example to strengthen it.

2. Line 102: Please elaborate what do you mean by coercive FP services in India?

METHOD:

3. In the materials and method section, please provide brief description about the study setting. For an international reader, it is difficult to know the context in which health services are being offered.

4. Line 122. Is it Indian financial year. If yes, please mention it.

5. The authors say design is described in Mozumdar et al. I think it would be appropriate for the authors to provide a gist of the method in this paper also.

6. Line 127: Was the facility manager/provider aware that women were being interviewed after the consultation? If Yes, I suggest authors to discuss how such prior knowledge would have influenced the study findings? Also, what steps were taken to reduce such bias?

observed?

7. Line 134: Authors suggest written informed consent was taken from all respondents? I am wondering how did you do that with women who could not read and write?

8. Line 173: Did you collect information on sex of the provider? If no, don't you think it is an important variable to consider.

9. Why the authors choose to use PCA not exploratory factor analysis. I feel this is the key weakness of this paper.

RESULTS:

10. Table 1: What is quality of counseling index? Did I miss the definition in the measure section?

11. Interesting you had more women with a male child. Can you clarify, if this is the sex of the last living child or something else? Also, given this biased distribution of women with male child, how it has an effect on the contraceptive use and quality of care received?

12. In Table 1, why provider characteristics are presented at patient level? They should presented at provider level, then only one can understand the distribution. I suggest rectifying this.

13. Line 218: There is nothing called confirmatory PCA. But, yes, there is confirmatory FA. I suggest authors to review the statistical analysis thoroughly.

14. Table 5: Instead of saying bivariate analysis, the author should present the title as "percent distribution of quality of counseling by....". Again, for providers, the analysis should at provider level by calculating the average of QoC at provider level

15. Table 6. If the focus of the paper is to understand relationship between QoC and contraceptive use, then Table 6 and 7 should be combined and only results for depicting this relationship should be presented. The rest of covariates, obviously, should be adjusted in the model. Presenting results for other covariates does not make much sense.

16. Table 6 and 7: Why authors did not use the categorical variable to test the association between QoC and contraceptive use? I strongly recommend the authors to remain consistent in use of measures across the paper.

DISCUSSION:

17. Line 307-314: While it is true that a fair share of women do not receive good QoC, the discussion should have focused in highlighting what are the structural, individual and provider level factor contributing it. Though some discussion is there in the subsequent paragraph, I feel more contextualization is required at the beginning itself.

18. Line 344-345. Can you elaborate more on how the study findings suggest that target-based family

planning without focus on QoC is a challenge? Challenge to what, how and to what extent?

19. Line 359-360: Authors recommend qualitative research on this issue. I feel this is very generic recommendation. It would be more useful to indicate what is the specific issue that needs more exploration. Already, several qualitative research is available exploring variable contraceptive use and non-use dynamics among women. What additional issues needs to be understand.

6. PLOS authors have the option to publish the peer review history of their article (what does this mean?). If published, this will include your full peer review and any attached files.

Reviewer #1: **Yes: **Sanjay K Mohanty

Reviewer #2: No

Reviewer #3: No

Reviewer #4: No

Reviewer #5: No

---

## [Author Response · Author response to Decision Letter 0]

16 Feb 2021

Response to reviewers

Additional Editor Comments

All five reviewers and I find that this paper has merit and can be considered if the authors address the reviewer comments. All five reviewers are well-respected scholars in the subject. Their comments range from a solid justification for the study at the introduction to giving more details on study design and to strengthen the statistical analyses and presentation of the tables. I believe many of these comments can be easily addressed. I hope to receive a revised version of this paper.

Journal requirements

2. Please include additional information regarding the survey or questionnaire used in the study and ensure that you have provided sufficient details that others could replicate the analyses. For instance, if you developed a questionnaire as part of this study and it is not under a copyright more restrictive than CC-BY, please include a copy, in both the original language and English, as Supporting Information. If the original language is written in non-Latin characters, for example Amharic, Chinese, or Korean, please use a file format that ensures these characters are visible.

We note that one or more of the authors are employed by a commercial company: Sambodhi Research and Communications.

(2) Please also provide an updated Competing Interests Statement declaring this commercial affiliation along with any other relevant declarations relating to employment, consultancy, patents, products in development, or marketed products, etc. 

Please respond by return email with an updated Funding Statement and Competing Interests Statement and we will change the online submission form on your behalf.

6. We note you have included a table to which you do not refer in the text of your manuscript. Please ensure that you refer to Table 2 in your text; if accepted, production will need this reference to link the reader to the Table.

The Authors would like to thank the editors and the reviewers for the time taken to provide insightful comments on the manuscript. We have addressed the Journal requirement by ensuring that the manuscript follows PLOS ONE’s style guidelines and added additional information including an amended Funding statement and an updated competing interest statement in the revised manuscript. We have also addressed the feedback from the reviewers and the response to their comments are described below:

Reviewer’s comments

Reviewer #1 comments

The motivation of paper is interesting nut the execution is weak. The paper need to improve in presentation and implications. The limitations of the study should also be highlighted. I put few of the point for consideration

Thank you for your consideration and feedback. We have addressed your comments in the revised manuscript and detail the way in which we have addressed these comments in the lines below. 

1. Introduction: It may be meaningful to begin with Quality of Care . Following this its linkages with contraceptive use. 

Based on this comment and comments from other reviewers, we have edited the Introduction substantially. The introduction now starts with a focus on QoC.

2.Review of literature Beyond Bruce framework is needed

To address this comment, we have referenced relevant work by Huezo and Diaz (1993) that focuses on meeting providers’ needs to ensure QoC and included Jain and Hardee’s work around revising the Bruce / Jain framework. 

3. Highlight why Uttar Pradesh and study sample

We have created a section called ‘Study Setting’ under Materials and Methods and explained the reason for selecting Uttar Pradesh and the study sample in that section. 

4. All tables need to be scientifically presented. I suggest you give 2 column , one for % and other for N. Moreover, many of the table are conbfusing to reader. For ex: table 1 n (%) or Mean (SD)? What is ecactly presented? Is it N and % or mean and SD. there are 2 values in tables. Most of the tables arein similar fashion

Thank you for this feedback. We have edited the presentation of all the tables in the manuscript, to make them clearer. Based on the guidance, we have created two separate columns to present n and % in all tables and clearly described cells where we present mean (std. dev.) or percentages. 

5. Drop table 2

Based on this comment, we have dropped table 2 from the manuscript and also replaced its mention from the text. 

6. Move table 3 to appendix

Based on this comment, we have moved table 3 to appendix. This table is now referred as Appendix A in the manuscript. 

7. OBC is not marginalised group as mentioned in text . Correct it

We recognize this oversight in our original text and have corrected it in the revised manuscript. 

8. Give mean value of composite score-QOFC

We have added the mean value of the composite QFPC score in line 291 in the revised manuscript (without track changes): 

9.Drop table 5. Too much segregation of small sample of about 80 is dangerous in a scientific manuscript.

We agree that the cell sizes become too small given the small sample size in the study. Based on this feedback we have replaced the bivariate analysis in Table 5 by adjusted linear regression with the QFPC score as the dependent variable. This analysis helps with criterion validity of the QFPC measure and avoids the issue of low cell-sizes of bi-variate analysis. The new table is referred as Table 4 in the revised manuscript. 

10. Add Strong note on limitations . the sampe is not true for generalisation. More over you are taking a selected sample. hecne do not generalise it

We concur that generalization is a challenge in our study sample. We have added text in the ‘strengths and limitations’ section to that effect and have modified the text in the discussion section to be cautious of generalization. 

11. Mention value addition of this and implications for policy

Based on this comment, we have added a new section to the manuscript ‘Program Implications’. This section discusses the implications of findings from this study in detail. 

Reviewer #2 comments

Reviewer #2: Overall, this paper has the potential to make an important contribution to the literature, particularly by pointing out the development and psychometric testing of a Quality of Family Planning Counseling. However, the sampling design and its description is not clear. The following are some of the comments.

Thank you for your detailed comments on the manuscript. We have addressed the comments in the revised manuscript as follows:

Abstract

The abstract needs editing.

The background section of abstract need to re-write. In the background section, it is written ad “This study involves the development and psychometric testing of a Quality of Family Planning Counseling (QFPC) measure for India….. I am not sure whether this statement is correct, as this analysis use a 237 women sample from public health facilities in Uttar Pradesh and generalised for India. Even there is no clear description on sample design in method section of abstract as well as in main methods. Whether the sampled public sector facilities spread across the state o

To address this comment, we have edited the abstract to clearly state that 120 public health facilities were sampled across Uttar Pradesh in India. We have also avoided implications that the measure is generalizable for India in the abstract. 

Background

I suggest that the authors say more accurately that the purpose of the analysis in the background section. There are no specific details on the family planning services or counciling in Uttar Pradesh. It is not clearly reading the background in connection with this study objective.

Based on this comment and comments from other reviewers, we have edited the Background substantially to make it more focused. We have also added context around Uttar Pradesh in the background as well as in the study settings.

Methods

Methods section is inadequate. The current method section in this manuscript reads in a very choppy way.

We have modified the methods section considerably to make the language smoother and more comprehensive. 

This section not explained how did the sample facilities selected from Uttar Pradesh, and how did the women sampled. I suggest that the authors say more accurately about the sampling design including public health facility selection, women selection etc.

Based on this comment, we have added a new section in Materials and Methods titled ‘Sampling Procedure’ that describes the sampling of facilities and women in detail.

Line no 123-124, it is mentioned that …. 178 facilities that met inclusion criteria, we sampled 120 facilities for study inclusion. Are these 178 facilities in the 75 districts in Uttar Pradesh or selected districts?

These facilities were from the 75 districts in Uttar Pradesh. We have modified the text to explicitly mention this point in the revised manuscript. 

Line 124-125, it is mentioned that “The design is described in Mozumdar et. al. (46)” Did you use the same data and analysed for Uttar Pradesh. It would be useful to explain the sample design in this manuscript as well. This will be helpful for the readers.

We have added text to describe the entire sampling design for the study without relying on Mozumdar et al. in the ‘Sampling Procedure’ section of the revised manuscript. 

Line 172-173, it is mentioned that “We additionally included measures on provider characteristics, captured through structured interviews with the specific providers who counseled women on FP services”. It would be useful to describe this clearly.

Thank you for this comment. Based on it, we have added a new section detailing the provider level covariates used in the study. This section is titled ‘Provider level covariates’ in the revised manuscript. 

Result

Did you collect the information on usage of family planning among women. If so it is useful to show the prevalence of family planning usage.

Yes, we had asked women if they used any modern method before and have included it as a dichotomous variable in our analysis. We also revised our multivariable models to include this variable in our manuscript. 

I could understand more than 60% of FP councillors were females as most of them where ANM or Nurses. Whether any male councillors were available in these facilities?

Thank you for this comment. A total of 144 providers counseled the N = 237 women in the sample. Of this, 15 providers were men. We have added this in the table as well as the text. The addition can be found in ‘Table 2’ and under the ‘Provider Characteristic’ section in the revised manuscript. In addition, we have included sex of the provider in our multivariable models and have revised our analyses accordingly. 

Table 7 can modify. Please show only odds ratio and 95% CI. For example, 1.05 (1.03-1.070. P value is not required as the odds with 95% CI is sufficient for interpreting the statistical significance.

We have modified table 7 substantially based on this comment and comments from other reviewers. We do not report the p-values in the revised table and have made other structural changes to the presentation of the table. The new table is referred as Table 5 in the revised manuscript. 

Discussion

In the discussion section, the sentence However, given the relevance of findings for public clinics, generalizability to this…………..." could be more explicit and mention clearly the reason. I am still doubtful, how the authors can generalize the result for India.

We concur that generalizing the findings from this study to India is a challenge. We have modified this in the discussion section and have been cautious not to generalize the findings to India. We have also added text to the Strengths and Limitations section in the revised manuscript to address this as a significant limitation of the study. 

Reviewer #3 comments

Reviewer #3: The paper titled “Measuring quality of family planning counseling and its effects on uptake of contraceptives in Public Health Facilities in Uttar Pradesh, India: a cross-sectional analysis” is a good attempt to examine the quality of contraceptive counselling and its subsequent effects on uptake of specific methods in Uttar Pradesh, India. The results are informative. Nevertheless, I suggest the followings to make the paper more suitable for publication:

Thank you for taking the time to review our manuscript. We have addressed your comments in the sections below:

Sampling design must be described in the paper, as the article is expected to stand alone; although it has been explained in other published article.

We have developed a new section under Materials and Methods titled ‘Sampling Procedure’ where we describe the sampling design for the study in detail. 

Does the survey captured the main motivator/factor that made sampled women to visit these facilities for contraception? Do these women come to the facility alone or with any family member or any grass root level health worker like ASHA/ANM? It is quite possible that these women might have been counselled for visiting these facilities for a particular method well in advance before visiting these facilities. How did the authors address this?

The survey did not include the main motivating factors for women to visit these facilities. It also did not capture any prior counseling received by these women from front line health workers like ASHAs or ANMs. We recognize this as a limitation of the study and added text in the Strengths and Limitations sections to reflect this. 

It is evidenced that usually most of the people seeking health care utilization from public health facilities are from low economic background. Hence inference drawn from this analysis has its own limitation so far as generalization is concerned. This may be marked in the discussion section. Does this survey gathered information about economic status of these women, if so inclusion of this in the analysis is suggested?

The survey did not include items to assess the economic status of the participants during the exit interview. This is a limitation of the study and we have added text in the Strengths and Limitations section to indicate the lack of this variable in our analysis.

It is suggested to discuss in details the possible reasons for those 13% women who preferred no method post-counselling; though they visited the facility for adopting/accepting any FP method.

To address this comment, we have added text that discusses the reasons for women not choosing any contraceptive post-counseling. This description can be found in lines 378 – 390 in the discussion section of the revised manuscript (without track changes).

Why the religion was classified as Muslim and Non-Muslims? It is assumed that majority of that Non-Muslims are Hindus, who constitute a majority of the sample women. In that case why not to classify Hindus and Others.

We categorized religion into Muslim and Non-Mulsim as modern contraceptive usage in Uttar Pradesh was disproportionately low among Muslim couples. We have added a justification for this classification in the measures section of the revised manuscript. 

The purpose behind capturing education up to primary level needs to be presented? Role of education may have been much better understood if it would have been collected beyond primary level.

Education was considered a confounding variable in the association between quality of counselling and contraceptive method used. Primary education was considered as a covariate based on the lower proportion of women in the state who complete higher education beyond primary.

The non-inclusion of the type of facility i.e. DH & CHC in the analysis, is unclear

We acknowledge the non-inclusion of this variable in the original manuscript and have revised our analyses to include the type of facilities in the revised manuscript. 

Table 7: It is not very clear OR at what level significance is shown/made bold.

We have removed all bold fonts from the tables and have reported 95% Cis to indicate level of significance.

Were there any non-literate participant? Especially when 35% of them had not completed primary education. If so, how did the consent sought and taken from them to participate in this study.

Yes, there were are few participants in the study who could not read or write. For such participants, the consent form was read out and in case they agreed to participate in the study, the interviewer sought their thumbprints as a proxy to their signature on the consent form.

The conclusion section must be revisited to limit it as per the analysis carried. Authors may restrict themselves from overgeneralization of the results looking at the sample size and design.

We have revised the conclusion section substantially based on this comment and comments from other reviewers. The discussion and conclusion in the revised manuscript avoids the pitfalls of overgeneralization. 

Reviewer #4 comments

Reviewer #4: No comments at this time.

No comments at this time.

No comments at this time.

No comments at this time.

No comments at this time.

No comments at this time.

No comments at this time.

No comments at this time.

Reviewer #5 comments

Reviewer #5: Thanks for giving me an opportunity to read this interesting paper. The authors have used a facility level exit interview data to present a topic which needs more and more research to improve quality of care for family planning services in India. While the paper has several strengths, I feel the statistical analysis is not that great. Following are observations which may help the authors in strengthening the paper.

Thank you for taking the time to review our manuscript and for sharing your comments on the paper. We have addressed your comments in the following ways:

INTRODUCTION

1. The introduction is too lengthy and I feel it can be shortened significantly. The authors should should strengthen the last but one paragraph in the introduction section (Page 6 Line 98-103) with more solid justification on why this study is called for. The current articulation is somewhat vague and may be supplemented with specific contextual example to strengthen it.

To address this comment, we have made substantive changes to the Introduction section that has made it much more comprehensive and has solid justification for the study.

2. Line 102: Please elaborate what do you mean by coercive FP services in India?

We have edited the introduction substantially and have dropped reference to the coercive FP services in India in the revised manuscript. 

METHOD:

3. In the materials and method section, please provide brief description about the study setting. For an international reader, it is difficult to know the context in which health services are being offered.

We have added text in the material and methods section to describe the study setting. 

4. Line 122. Is it Indian financial year. If yes, please mention it.

Yes, we were referring to the Indian Financial year. We have added text to clarify that in the revised manuscript. 

5. The authors say design is described in Mozumdar et al. I think it would be appropriate for the authors to provide a gist of the method in this paper also.

Based on this comment and similar comments from other reviewers, we have added a new section titled ‘Sampling Procedure’ that describe the study design and does not rely on Mozumdar et al for the description. 

6. Line 127: Was the facility manager/provider aware that women were being interviewed after the consultation? If Yes, I suggest authors to discuss how such prior knowledge would have influenced the study findings? Also, what steps were taken to reduce such bias?

observed?

Yes, the facility in-charge were informed about the interviews being conducted. However, the providers were not aware of the clients being interviewed after the consultation. Care was taken by the survey team to ensure that the post-consultation interviews were conducted away from the vicinity of the place where the counseling services were being provided.

7. Line 134: Authors suggest written informed consent was taken from all respondents? I am wondering how did you do that with women who could not read and write?

There were a few respondents who could not read or write. In such cases, the consent form was read to them and if they agreed to participate in the interview, their thumbprints were taken as a proxy to their signatures on the consent form. 

8. Line 173: Did you collect information on sex of the provider? If no, don't you think it is an important variable to consider.

Yes, we collected information on sex of providers. Out of a total of 144 providers who counseled the clients sampled in the study, 15 were males. We have included a variable on sex of provider in the analysis in our revised manuscript. 

9. Why the authors choose to use PCA not exploratory factor analysis. I feel this is the key weakness of this paper.

We used PCA because one of our objective was to create a composite score of quality of Family Planning counseling from our survey items. PCA allowed us to create a linear combination of these items and create the QFPC measure, which is the outcome of interest in our study.

RESULTS:

10. Table 1: What is quality of counseling index? Did I miss the definition in the measure section?

Thank you for pointing this oversight. We were referring to the Quality of Family Planning Counseling (QFPC) measure. We have rectified this in the revised manuscript. 

11. Interesting you had more women with a male child. Can you clarify, if this is the sex of the last living child or something else? Also, given this biased distribution of women with male child, how it has an effect on the contraceptive use and quality of care received?

Thank you for this comment. The variable on presence of a male child identifies women who have at least 1 male child. This male child can have any birth order and is not specific to the last birth. We included this variable as a confounder between our dependent and independent variables and adjusted for it in all our analyses. 

12. In Table 1, why provider characteristics are presented at patient level? They should presented at provider level, then only one can understand the distribution. I suggest rectifying this.

We agree that providing provider level characteristics separately would clearly describe the distribution to readers. Based on this comment, we have created a new table that contains information on characteristics of providers. This new table is referred as Table 2 in the revised manuscript. 

13. Line 218: There is nothing called confirmatory PCA. But, yes, there is confirmatory FA. I suggest authors to review the statistical analysis thoroughly.

We acknowledge this oversight and have corrected the wordings in the text to reflect our analysis accurately.

14. Table 5: Instead of saying bivariate analysis, the author should present the title as "percent distribution of quality of counseling by....". Again, for providers, the analysis should at provider level by calculating the average of QoC at provider level

We have changed the analysis in the paper substantially. The purpose of table 5 was to assess criterion validity for the QFPC measure. Instead of undertaking a bivariate analysis, which suffered form low cell sizes, we developed adjusted linear regression models for the purpose, which are more robust. The new table is referred as Table 4 in the revised manuscript. 

15. Table 6. If the focus of the paper is to understand relationship between QoC and contraceptive use, then Table 6 and 7 should be combined and only results for depicting this relationship should be presented. The rest of covariates, obviously, should be adjusted in the model. Presenting results for other covariates does not make much sense.

Based on this comment and comments from other reviewers, we have modified the analysis in the paper substantially. For all our multivariable models, we only present the coefficients for the primary independent variables and adjust for the rest of the covariates as suggested. 

16. Table 6 and 7: Why authors did not use the categorical variable to test the association between QoC and contraceptive use? I strongly recommend the authors to remain consistent in use of measures across the paper.

Based on this comment, we have ensured consistency in the use of the QFPC measure across the paper. In order to preserve the information in the data, we chose to use the QFPC measures as a continuous variable throughout the paper. This revision is reflected across all the revised tables and the results section in the revised manuscript. 

DISCUSSION:

17. Line 307-314: While it is true that a fair share of women do not receive good QoC, the discussion should have focused in highlighting what are the structural, individual and provider level factor contributing it. Though some discussion is there in the subsequent paragraph, I feel more contextualization is required at the beginning itself.

The initial two paragraphs in the discussion section are focused on reliability and validity of the QFPC measure and placing it in the context of other similar studies. To address this comment, we have edited the third paragraph of the discussion section (lines 345-354 in the revised manuscript) to focus on the structural factors associated with Quality of Counselling. 

18. Line 344-345. Can you elaborate more on how the study findings suggest that target-based family

planning without focus on QoC is a challenge? Challenge to what, how and to what extent?

We recognize that this statement was a bit out of context from our main discussion point. We have removed this and enhanced focus on the quality and quantity of training to providers in the revised manuscript. 

19. Line 359-360: Authors recommend qualitative research on this issue. I feel this is very generic recommendation. It would be more useful to indicate what is the specific issue that needs more exploration. Already, several qualitative research is available exploring variable contraceptive use and non-use dynamics among women. What additional issues needs to be understand.

To address this comment, we have removed the generic recommendation for Qualitative research and call for further research to understand the reasons behind women arriving at facilities for FP methods, but leaving without selecting any method.

---

## [Decision Letter · Decision Letter 1]

12 Apr 2021

Measuring quality of family planning counseling and its effects on uptake of contraceptives in Public Health Facilities in Uttar Pradesh, India: a cross-sectional analysis

PONE-D-20-28620R1

Dear Dr. Dey,

We’re pleased to inform you that your manuscript has been judged scientifically suitable for publication and will be formally accepted for publication once it meets all outstanding technical requirements.

Kind regards,

Srinivas Goli, Ph.D.

Academic Editor

PLOS ONE

Additional Editor Comments (optional):

Authors have effectively addressed comments of multiple reviewers. In its current form, this paper can be accepted for publication in PLOS.

Reviewers' comments:

Reviewer's Responses to Questions

**Comments to the Author**

1. If the authors have adequately addressed your comments raised in a previous round of review and you feel that this manuscript is now acceptable for publication, you may indicate that here to bypass the “Comments to the Author” section, enter your conflict of interest statement in the “Confidential to Editor” section, and submit your "Accept" recommendation.

Reviewer #3: All comments have been addressed

2. Is the manuscript technically sound, and do the data support the conclusions?

Reviewer #3: Yes

3. Has the statistical analysis been performed appropriately and rigorously? 

Reviewer #3: Yes

4. Have the authors made all data underlying the findings in their manuscript fully available?

Reviewer #3: No

5. Is the manuscript presented in an intelligible fashion and written in standard English?

Reviewer #3: Yes

6. Review Comments to the Author

Reviewer #3: Thank you for revising the paper based on the comments. This draft is much focused and deemed to add to the Family Planning evidence base.

7. PLOS authors have the option to publish the peer review history of their article (what does this mean?). If published, this will include your full peer review and any attached files.

Reviewer #3: **Yes: **Manas Ranjan Pradhan

---

## [Editor Report · Acceptance letter]

23 Apr 2021

PONE-D-20-28620R1 

Measuring quality of family planning counselling and its effects on uptake of contraceptives in Public Health Facilities in Uttar Pradesh, India: a cross-sectional analysis 

Dear Dr. Dey:

I'm pleased to inform you that your manuscript has been deemed suitable for publication in PLOS ONE. Congratulations! Your manuscript is now with our production department. 

Kind regards, 

on behalf of

Dr. Srinivas Goli 

Academic Editor

PLOS ONE